# Genotyping of familial Mediterranean fever gene (*MEFV*)—Single nucleotide polymorphism—Comparison of Nanopore with conventional Sanger sequencing

Jonas Schmidt[1,2,3], Sandro Berghaus[1], Frithjof Blessing[1,2], Holger Herbeck[1], Josef Blessing[1], Peter Schierack[3,4], Stefan Rödiger[3,4], Dirk Roggenbuck [ID][3,4☯]*, Folker Wenzel[2☯]

1 Institute for Laboratory Medicine, Singen, Germany, 2 Faculty of Medical and Life Sciences, Furtwangen University, Villingen-Schwenningen, Germany, 3 Faculty Environment and Natural Sciences, Institute of Biotechnology, Brandenburg University of Technology Cottbus-Senftenberg, Senftenberg, Germany, 4 Faculty of Health Sciences Brandenburg, Brandenburg University of Technology Cottbus–Senftenberg, Senftenberg, Germany

☯ These authors contributed equally to this work.
* dirk.roggenbuck@b-tu.de

**Data Availability Statement:** The Nanopore sequencing data underlying the results presented in the study are available from the European

## Abstract

### Background

Through continuous innovation and improvement, Nanopore sequencing has become a powerful technology. Because of its fast processing time, low cost, and ability to generate long reads, this sequencing technique would be particularly suitable for clinical diagnostics. However, its raw data accuracy is inferior in contrast to other sequencing technologies. This constraint still results in limited use of Nanopore sequencing in the field of clinical diagnostics and requires further validation and IVD certification.

### Methods

We evaluated the performance of latest Nanopore sequencing in combination with a dedicated data-analysis pipeline for single nucleotide polymorphism (SNP) genotyping of the familial Mediterranean fever gene (*MEFV*) by amplicon sequencing of 47 clinical samples. Mutations in *MEFV* are associated with Mediterranean fever, a hereditary periodic fever syndrome. Conventional Sanger sequencing, which is commonly applied in clinical genetic diagnostics, was used as a reference method.

### Results

Nanopore sequencing enabled the sequencing of 10 target regions within *MEFV* with high read depth (median read depth 7565x) in all samples and identified a total of 435 SNPs in the whole sample collective, of which 29 were unique. Comparison of both sequencing workflows showed a near perfect agreement with no false negative calls. Precision, Recall, and F1-Score of the Nanopore sequencing workflow were > 0.99, respectively.

Nucleotide Archive (Accession: PRJEB49157). The data analysis pipeline as well as the Sanger sequencing results are available from Github (github.com/j4yo/MEFV-SNP-Genotyping-Pipeline).

**Funding:** The authors received no specific funding for this work.

**Competing interests:** The authors have declared that no competing interests exist.

## Conclusions

These results demonstrated the great potential of current Nanopore sequencing for application in clinical diagnostics, at least for SNP genotyping by amplicon sequencing. Other more complex applications, especially structural variant identification, require further in-depth clinical validation.

## 1. Introduction

Since its first description in 1996, nanopore-based deoxyribonucleic acid (DNA) sequencing has developed to one of the most powerful sequencing technologies thanks to continuous innovation and improvements [1,2]. Nowadays, different sequencing devices and protocols are commercially available rendering this technique attractive for various areas of molecular biological research and diagnostics, including metagenomics, bacterial and viral infectiology, human genomics, and cancer research [3–11]. The core components of current Nanopore sequencing devices are protein nanopores contained in a membrane [12,13]. As single DNA molecules are passed through these pores, the resulting changes in an ionic current across the membrane are used to infer the sequence of nucleic acids [11–13]. This sequencing approach offers the advantages of real-time sequencing, ultra-long read length (average read length up to 10 kb), high throughput and the possibility of base modification detection as well as native ribonucleic acid (RNA) sequencing [1,13,14]. However, a major drawback compared to other next-generation sequencing (NGS) techniques has been the comparatively high error rate [13]. Although this is a heterogenous measure, which is influenced by different parameters including sequencing instrument, sequencing protocol and sample type, Nanopore sequencing shows a distinct higher error rate (~6%) compared to PacBio sequencing (~1.5%), Illumina sequencing (~0.5%) and conventional Sanger sequencing (~0.001%) [15–19]. This is especially critical for medical applications such as single nucleotide polymorphism (SNP) genotyping, which require high sequencing accuracy to achieve reliable results [13]. Although the accuracy of Nanopore sequencing has improved considerably by optimization of the underlying sequencing chemistry and bioinformatic analysis tools, it is important to validate the technique against established gold standard methods such as Sanger sequencing to assess a possible application in medical diagnostics [13,20].

A common monogenetic autoinflammatory disease is Familial Mediterranean fever (FMF) which shows a high prevalence among Turkish, Armenian, Jewish and Arabic communities from the eastern Mediterranean region [21,22]. The disease is a clinical diagnosis and mainly characterized by recurrent fever and serositis, with amyloidosis being a severe complication in untreated individuals [22–24]. FMF is considered to be inherited autosomal recessive and is associated with point mutations (single substitutions) in the Mediterranean Fever (*MEFV*) gene [22,24]. This gene consists of 10 exons and is located on the short arm of chromosome 16 in minus strand orientation [22]. It encodes a 781 amino acids containing protein called pyrin, which plays a key role in apoptosis and inflammatory pathways. It is mainly expressed in neutrophils, eosinophils, dendritic cells and fibroblasts [21–23]. Mutated pyrin is thought to cause an excessive inflammatory response through uncontrolled interleukin-1 (IL-1) secretion [21,25]. After clinical diagnosis, the diseases is generally treated with colchicine, and IL-1 blockade is suggested in refractory cases [21]. Genetic testing is employed to aid in the clinical diagnosis of FMF and to screen relatives at risk [23]. This can be done either by testing for the most common mutations (targeted mutation analysis) or by sequencing of selected exons [23]. According to expert consensus guidelines for the genetic diagnosis of hereditary recurrent fevers a minimum diagnostic screen should include clearly pathogenic variants which are

frequently identified in patients [26]. For FMF this incorporates the exons 2, 3, 5 and 10 of *MEFV* or a set of nine variants [26]. While DNA sequencing is used in most laboratories for variant analysis, targeted approaches can also be applied by using PCR based or reverse-hybridization based assays [26]. However, these targeted approaches as well as conventional Sanger sequencing suffer from the technological limitation that only a comparably small genetic target range can be covered within a single run. To overcome this limitation, NGS can be applied to sequence gene panels including not only *MEFV* for the diagnosis of FMF but also genes which are associated with other periodic fever syndromes like mevalonate kinase deficiency (MKD, gene *MVK*), tumor necrosis factor receptor-associated periodic syndrome (TRAPS, gene *TNFRSF1A*) and cryopyrin-associated periodic syndrome (CAPS, gene NLRP3) [26,27].

In this study, to evaluate the clinical performance of current Nanopore sequencing, we applied this sequencing technique in combination with a dedicated data analysis pipeline for SNP genotyping of selected regions of *MEFV* in 47 patients and validated the results against diagnostic Sanger sequencing as the gold standard method.

## 2. Material and methods

### 2.1 Clinical samples

Samples from 25 female and 22 male patients that were drawn for routine *MEFV* assessment were included into this study after routine testing by Sanger sequencing was performed. Median age was 12.1 years (interquartile range [IQR] 12.9). Primary blood samples were collected in EDTA collection tubes by venipuncture and stored at 4°C until further processing. The routine diagnostic workflow includes DNA isolation, polymerase chain reaction (PCR) amplification of selected targets within *MEFV* and Sanger sequencing as described below. Subsequent to routine Sanger sequencing, the amplicons obtained from the amplification step were pooled per sample and Nanopore sequencing was performed.

All included individuals gave their written informed consent. For minor patients, written informed consent was obtained from the parents. The study followed all relevant national regulations and institutional policies, has been approved by the ethics committee of the Landesärztekammer Baden-Württemberg (F-2018-089) and complies with the World Medical Association Declaration of Helsinki regarding ethical conduct of research involving human subjects and/or animals.

### 2.2 DNA isolation and PCR amplification

DNA isolation from EDTA whole blood samples was performed on chemagic Prepito-D instruments (PerkinElmer, Waltham, USA) using Prepito NA Body Fluid kits (PerkinElmer) (expected yield: ~2.5 μg).

PCR amplification of the *MEFV* target regions was performed stepwise in eight different PCR reactions using target specific primers (Biomers, Ulm, Germany), Q-Solution (Qiagen, Hilden, Germany), and the AmpliTaq Gold 360 Master Mix (ThermoFisher Scientific, Waltham, USA). The amplicons were designed to span *MEFV* exon 1, exon 2, exon 3, exon 4, exon 5, exon 6, exon 7/8, and exon 9/10 (S1 Table). PCR reactions were performed on an Applied Biosystems Veriti thermal cycler (ThermoFisher Scientific) (S2 and S3 Tables). Nuclease free water was included in all runs as a no template control.

### 2.3 Sanger sequencing

Prior to sequencing, a clean-up of the amplicons was performed by using ExoSAP-IT clean-up kits (ThermoFisher Scientific). Briefly, 7 μL PCR product were mixed with 1 μL clean-up

reagent by pipetting. This reaction mix was incubated for 15 min at 37˚C followed by 15 min at 80˚C.

Sanger sequencing of the purified amplicons was performed using the BigDye Terminator Version 3.1 kit (ThermoFisher Scientific) on an Applied Biosystems 3500 Dx Series Genetic Analyzer (ThermoFisher Scientific) according to the manufacturer's protocol. Briefly, sequencing reactions were set up using target specific sequencing primers (Biomers) (S4 Table). After incubation on a thermal cycler, the reaction mix was cleaned by precipitation with ethanol/EDTA/sodium acetate and loaded on the instrument for capillary electrophoresis after resuspending in injection buffer. Sequencing was performed using POP-6 Polymer (ThermoFisher Scientific).

## 2.4 Nanopore sequencing

Prior to Nanopore sequencing, equal volumes (10 μL) of the amplicons from the target amplification step were pooled for each individual sample. DNA concentration of the pooled samples was measured on a Qubit 4 fluorometer (ThermoFisher Scientific) using the 1x dsDNA HS assay (ThermoFisher Scientific) (S5 Table). Afterwards, a 1.8x AMPure XP bead clean-up was performed according to the manufacturer's protocol (Beckman Coulter, Brea, USA). Sequencing libraries were prepared according to the manufacturer's protocol using native barcoding kits (EXP-NBD104, EXP-NBD114) in combination with ligation sequencing kits (SQK-LSK109) (Oxford Nanopore Technologies (ONT), Oxford, UK). The libraries were prepared with a total of 12 samples per library for each run to ensure a sufficient read count per sample and that the relative proportion of a single sample is comparable (S5 Table). DNA input per sample was 200fmol and 12.5fmol of each barcoded sample were pooled prior to sequencing. Sequencing was performed on a MinION sequencing device (ONT) for 6h using R9.4.1 flow cells (ONT). All samples were sequenced in four different runs using two flow cells. Prior to reuse, the flow cells were purged according to the manufacturer's protocol using flow cell wash kits (EXP-WSH003) (ONT).

## 2.5 Sequencing data analysis

Sanger sequencing data was analyzed using SEQUENCE Pilot Software [v 3.4.2] (JSI medical systems GmbH, Ettenheim, Germany). Variants were called against the *MEFV* reference (ENSEMBL gene: ENSG00000103313; transcript: ENST00000219596). Identified variants were manually inspected and exported to a comma separated-values (csv) file for comparison with the Nanopore sequencing results.

To analyze the Nanopore sequencing data, a dedicated data analysis pipeline was established by us and implemented into a bash shell script for automation purpose (Fig 1). Raw data in FAST5 file format was basecalled and demultiplexed using the Guppy Basecalling Software [v 5.0.11+2b6dbffa5] (ONT). Basecalling was performed using the "super-accurate" basecalling model (dna_r9.4.1_450bps_sup.cfg). Basic run quality control was performed by applying pycoQC [v 2.5.2] (github.com/tleonardi/pycoQC). To remove chimeric and low-quality reads, read filtering was done with NanoFilt [v 2.7.1] (github.com/wdecoster/nanofilt). The filter was set to keep only reads with a read length between 250 and 1200 bases and a quality score equal or larger 15. After filtering, the reads were aligned to chromosome 16 of the hg19 reference genome (NC_000016.9) using minimap2 [v 2.20-r1061] (github.com/lh3/minimap2). The resulting Sequence Alignment Map (SAM) files were sorted and indexed with Samtools [v 1.7] (github. com/samtools/samtools). Afterwards, bcftools [v 1.13] (github.com/samtools/bcftools) was used for variant calling. The tool was set to include only SNPs and skip insertions and deletions. Variant filtering was performed by applying bedtools [v 2.30.0] (github.com/arq5x/bedtools2). Only

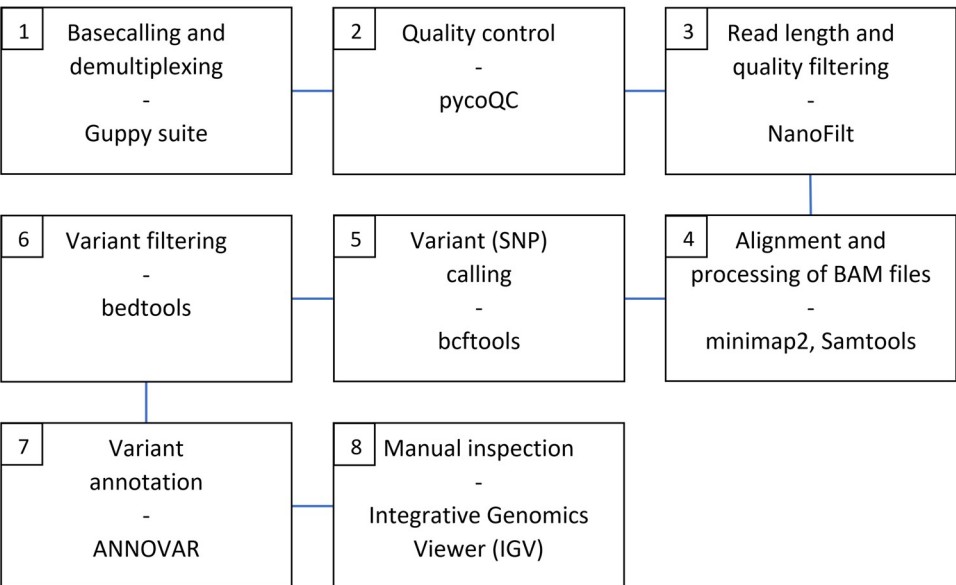

**Fig 1. Data analysis pipeline applied for the assessment of the Nanopore sequencing data.** Tools used for the different tasks are shown. Step 1 to 7 were implemented in a bash shell script for automation purpose. SNP; single nucleotide polymorphism.

calls in *MEFV* regions covered by the amplicons were included into the final data set. Finally, the identified variants were annotated using ANNOVAR [v 2018-04-16] [28].

Once the automated data analysis pipeline was complete, the results for each individual sample were manually reviewed using the Integrative Genomics Viewer [v2.10.3] (github.com/igvteam/igv).

## 2.6 Results comparison

Method comparison was done in R [v 3.6.3] (R Foundation for Statistical Computing, Vienna, Austria) [29]. After importing the data sets, Nanopore sequencing variant calls were compared to the Sanger sequencing reference for genomic position, nucleotide change, zygosity, amino acid position, and amino acid change. Nanopore sequencing calls were only classified as true positive (TP) if all five criteria matched to the corresponding Sanger sequencing reference. Variants without a complete match as well as variants which were missed by Nanopore sequencing were classified as false negative (FN) and variants, which were solely identified by Nanopore sequencing as false positive (FP). Based on these classifications, comparative measures including Precision (TP/(TP + FP)), Recall (TP/(FN + TP)) and F1-Score (2 * (Precision * Recall)/(Precision + Recall)) were calculated [30].

Data visualization was performed in R as well using the packages ggVennDiagram, ggplot2, gggenes, and ggpubr. Sequencing depth information was extracted from the SAM files prior to visualization using Samtools.

## 3. Results

To evaluate the performance of Nanopore sequencing for SNP genotyping, we performed amplicon sequencing of selected *MEFV* regions in 47 clinical samples using a MinION sequencing device and compared the results to conventional Sanger sequencing.

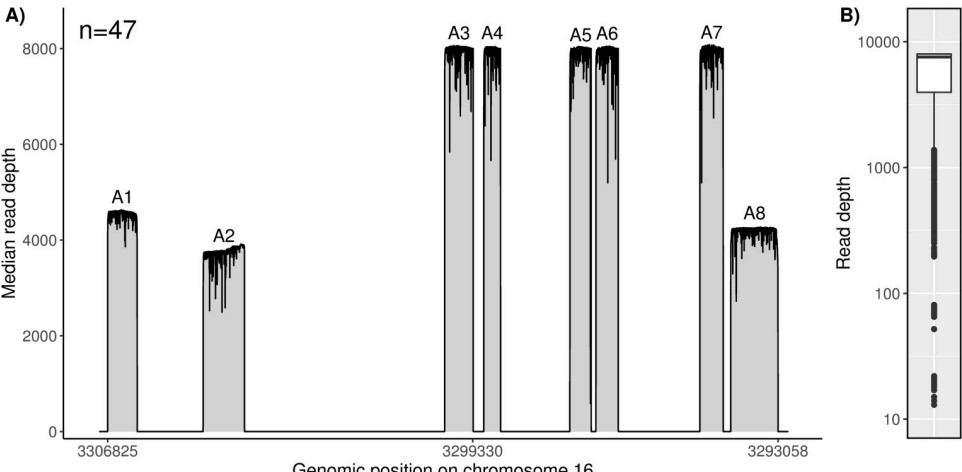

**Fig 2. Visualization of the read depth distribution achieved by Nanopore sequencing.** (A) Median read depth achieved by amplicon sequencing of selected regions in the *MEFV* gene in 47 clinical samples using a MinION sequencing device. The target regions cover the relevant regions of all 10 exons of this gene. (B) Read depth distribution in the target regions over all 47 samples. A median read depth of 7565x (IQR 4025) was achieved. Outliers with a reduced sequencing depth were observed at the edges of individual amplicons.

By using Nanopore sequencing in combination with a dedicated data analysis pipeline, it was possible to sequence the eight amplicons covering the relevant *MEFV* regions of all 10 exons with a median read depth of 7565x (IQR 4025) over all 47 samples (Fig 2B). A reduced read depth was observed at the edges of individual amplicons (minimum 13x). Furthermore, differences in the median read depth between different amplicons were observed (Fig 2A). Overall, amplicon 1, 2, and 8 showed a lower median read depth compared to the remaining amplicons.

In total, 433 SNPs were identified in the investigated sample collective by Sanger sequencing (284 heterozygous and 149 homozygous). They include 28 unique variants of which 13 are non-synonymous (Table 1). The most common non-synonymous variants include p.E148Q (40.4%), p.R202Q (34.0%), p.M694V (25.5%), p.P369S (12.8%) and p.R408Q (12.8%). In addition, the most common synonymous variants were p.R314R (76.6%), p.E474E (70.2%), p. Q476Q (70.2%), p.D510D (70.2%), and p.P588P (68.1%).

All 433 SNPs confirmed by Sanger sequencing in the sample collective were also identified by Nanopore sequencing with matching genomic position, nucleotide change, zygosity, amino acid position, and amino acid change (Fig 3). Additionally, the Nanopore sequencing results showed a transversion from guanine (G) to thymine (T) in the 3' untranslated region (UTR) at genomic position 3293090 in two patients which has not been identified by initial Sanger sequencing (Figs 3 and S1). Read depth at this genomic position was >7000x in both cases. A data base research, including ClinVar and dbSNP, did not reveal any further information on this SNP. Remarkably, both individuals in whom this SNP was identified were related. By sequencing an additional amplicon, spanning this region, it was possible to confirm the transversion in both samples also by Sanger sequencing (S2 Fig).

For further method comparison, performance parameters such as Precision, Recall, and F1-Score were calculated from the results. The SNP which was only identified by Nanopore sequencing was treated as false positive, since it was not identified during the initial diagnostic Sanger sequencing runs. Based on this assumption, the Nanopore sequencing method in comparison to Sanger sequencing showed a Precision of 0.995, a Recall of 1 and a F1-Score of 0.998.

**Table 1. Unique *MEFV* variants identified in 47 patients.** Variant frequency in the sample collective under investigation is shown. One variant in two patients was only identified by Nanopore sequencing and could not be confirmed by initial Sanger sequencing.

| Genomic position[a] | cDNA[b] | Protein[c] | Region | Exon[c] | Count (%) | Function[d] | Agreement[e] |
|---|---|---|---|---|---|---|---|
| 3299749 | c.942C>T | p.R314R | exonic | 3 | 36 (76.6) | S | yes |
| 3298865 | rs224212 | - | intronic | - | 33 (70.2) | - | yes |
| 3297181 | c.1422G>A | p.E474E | exonic | 5 | 33 (70.2) | S | yes |
| 3297175 | c.1428A>G | p.Q476Q | exonic | 5 | 33 (70.2) | S | yes |
| 3297073 | c.1530T>C | p.D510D | exonic | 5 | 33 (70.2) | S | yes |
| 3293888 | c.1764G>A | p.P588P | exonic | 9 | 32 (68.1) | S | yes |
| 3293922 | rs1231123 | - | intronic | - | 30 (63.8) | - | yes |
| 3296616 | rs224205 | - | intronic | - | 29 (61.7) | - | yes |
| 3296429 | rs224204 | - | intronic | - | 29 (61.7) | - | yes |
| 3304762 | c.306T>C | p.D102D | exonic | 2 | 21 (44.7) | S | yes |
| 3304654 | c.414A>G | p.G138G | exonic | 2 | 21 (44.7) | S | yes |
| 3304573 | c.495C>A | p.A165A | exonic | 2 | 21 (44.7) | S | yes |
| 3304626 | c.442G>C | p.E148Q | exonic | 2 | 19 (40.4) | NS | yes |
| 3304463 | c.605G>A | p.R202Q | exonic | 2 | 16 (34.0) | NS | yes |
| 3293407 | c.2080A>G | p.M694V | exonic | 10 | 12 (25.5) | NS | yes |
| 3299586 | c.1105C>T | p.P369S | exonic | 3 | 6 (12.8) | NS | yes |
| 3299468 | c.1223G>A | p.R408Q | exonic | 3 | 6 (12.8) | NS | yes |
| 3293310 | c.2177T>C | p.V726A | exonic | 10 | 4 (8.5) | NS | yes |
| 3297100 | c.1503C>T | p.R501R | exonic | 5 | 3 (6.4) | S | yes |
| 3294246 | rs77380520 | - | intronic | - | 3 (6.4) | - | yes |
| 3293257 | c.2230G>T | p.A744S | exonic | 10 | 3 (6.4) | NS | yes |
| 3293205 | c.2282G>A | p.R761H | exonic | 10 | 3 (6.4) | NS | yes |
| 3293403 | c.2084A>G | p.K695R | exonic | 10 | 2 (4.3) | NS | yes |
| 3293090 | - | - | UTR3 | - | 2 (4.3) | - | no |
| 3304380 | c.688G>A | p.E230K | exonic | 2 | 1 (2.1) | NS | yes |
| 3304317 | c.751G>A | p.E251K | exonic | 2 | 1 (2.1) | NS | yes |
| 3304158 | c.910G>A | p.G304R | exonic | 2 | 1 (2.1) | NS | yes |
| 3293447 | c.2040G>C | p.M680I | exonic | 10 | 1 (2.1) | NS | yes |
| 3293369 | c.2118G>A | p.P706P | exonic | 10 | 1 (2.1) | S | yes |

[a]Genomic position on the hg19 reference genome (NC_000016.9).

[b]dbSNP identifiers are shown for variants in non-coding regions.

[c]Amino acid information and exon number are only shown for variants in exonic regions.

[d]S = synonymous; NS = non-synonymous.

[e]Agreement between Nanopore sequencing and initial Sanger sequencing results.

## 4. Discussion

To evaluate the performance of Nanopore sequencing for SNP genotyping by amplicon sequencing, we performed a comprehensive method comparison with conventional Sanger sequencing using 47 clinical samples from patients with suspicion of FMF. The number of studies comparing Nanopore and Sanger sequencing in diagnostics has been limited [31–33]. Routine diagnostics using Sanger sequencing, the current gold standard for point-mutation detection so far, revealed the presence of various SNPs, including the non-synonymous variants p.E148Q, p.R202Q, p.M694V, p.P369S and p.R408Q in this sample collective [34]. All of these mutations have been previously described in FMF patients [22,35].

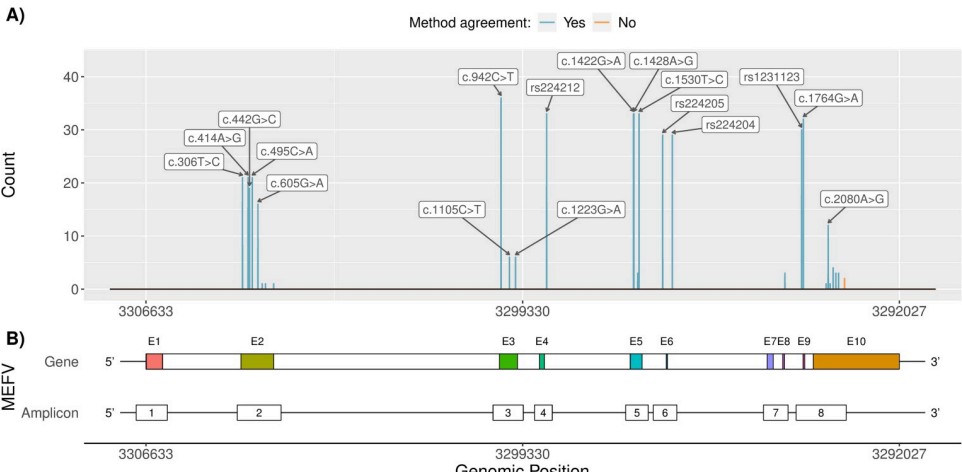

**Fig 3. Genetic variants which were identified in selected regions of *MEFV*.** (A) Frequency of single nucleotide polymorphisms (SNPs) identified in 47 clinical samples by Sanger and Nanopore sequencing. cDNA labels or dbSNP references are given for the most common variants. Variants with a complete agreement between Sanger and Nanopore sequencing in all 47 clinical samples are coloured in blue and differing variants are coloured in orange. (B) Gene map of *MEFV* and the amplicons used to sequence selected regions of this gene (S1 Table). Genomic positions on the hg19 reference genome (NC_000016.9) are shown in minus strand orientation.

By performing Nanopore sequencing on a MinION sequencing device in combination with a dedicated data analysis pipeline, it was possible to sequence the relevant regions of all *MEFV* exons with a very high read depth. All variants previously identified by diagnostic Sanger sequencing were also accurately detected. Furthermore, Nanopore sequencing revealed only one SNP in two related patients, which had not been identified during initial Sanger sequencing. This SNP was located in the 3' UTR at the edge of the amplicon covering this region. Since current Sanger sequencing is based on PCR amplification and capillary electrophoresis, poor sequence quality due to primer binding and insufficient base resolution is a very common problem at the beginning and end of an individual read [36]. Therefore, low-quality regions are trimmed prior to data analysis. For this reason, the diverging SNP is located in a region of amplicon 8, which cannot be properly sequenced by Sanger sequencing on either the forward or reverse strand. In Nanopore sequencing, a similar problem does not occur since the sequencing adapters are ligated to the ends of the PCR products during library preparation [37]. By sequencing an additional amplicon, spanning the relevant region of the 3' UTR, we were able to confirm the transversion in both patients also by Sanger sequencing. Taking these additional results into account, our data show a complete agreement between Nanopore and Sanger sequencing. Nevertheless, a comprehensive data-base research did not reveal any information about the clinical relevance of this transversion. Since the initial diagnostic Sanger sequencing runs did not identify this variant, the corresponding variant calls were treated as false positive in the calculation of performance measures.

The obtained Precision, Recall, and F1-Score of > 0.99 each demonstrate the excellent agreement between Nanopore and Sanger sequencing for SNP genotyping in our study [38]. This is consistent with other studies that also reported a high degree of agreement for various applications, especially in microbiology and cancer genomics [31,39–41].

The limitations of our study were the small sample size and the focus on targeted SNP genotyping alone. By using targeted amplicon sequencing on the MinION, we were able to sequence the relevant regions of the *MEFV* exons at a high read depth (median read depth 7565x). However, there is a substantial amount of variation in read depth between different

**Table 2. Comparison of Nanopore and Sanger sequencing based on various aspects relevant for use in clinical diagnostics.**

| Aspect | Sanger sequencing | Nanopore sequencing |
|---|---|---|
| Capital costs (Instrument, Computing unit, Software)[a] | High (~200000 €) | Low (~3500 €) |
| Price per *MEFV* sample [€][b] | 160 | 75 |
| Time to result [workdays][c] | 3 | 3 |
| Multiplexing | No | Yes |
| Data analysis | Simple | Complex |
| Application in clinical genetics | Reference method | Validation needed |

[a]Based on current list prices.

[b]Approximate price per sample. To archive highest diagnostic accuracy, 11 sequencing reactions must be performed to sequence all target regions with Sanger sequencing, since amplicon 2 and 8 are sequenced in three and two sequencing reactions, respectively. For Nanopore sequencing, the price decreases with increasing degree of multiplexing. [c]Includes DNA isolation, PCR amplification, sequencing and data analysis.

amplicons within one sample and different samples. This was based on the varying DNA input and varying efficacy of the eight PCR reactions used to amplify the *MEFV* target regions. A more homogeneous read depth distribution could be achieved by determining the concentration of the individual amplicons prior to pooling and subsequent pooling of equimolar amounts. Although this would increase the complexity of the protocol, it would contribute to more homogenous results and probably facilitate a higher degree of multiplexing. Multiplexing of different clinical samples is a key factor in diagnostic NGS as it significantly improves cost efficiency (Table 2) [31]. According to Leija-Salazar et al. a read depth of >100x could be sufficient for accurate variant identification by Nanopore sequencing [10]. Such a threshold would remarkably increase the possible degree of multiplexing in our experimental design. However, due to the inhomogeneous read depth distribution between different amplicons we were not able to evaluate this accurately by subsampling of the data.

Due to the high read depth achieved by amplicon sequencing, we were able to use bcftools for accurate variant calling. This tool employs Bayesian statistics to determine the most likely genotype [38,42]. However, modern diagnostic NGS applications mainly involve gene panel sequencing, whole exome sequencing, and whole genome sequencing [38]. Due to the obviously larger target space, the median read depth in such applications is normally much lower than in amplicon sequencing. Therefore, under these circumstances, it may be necessary to apply more modern tools for accurate variant calling, such as Nanopolish and Medaka (github.com/nanoporetech/medaka), that can handle the unique Nanopore sequencing error profile even at low read depth [43]. Further, structural variant calling including deletions, inversions, tandem duplications, insertions, transpositions, and translocations from Nanopore sequencing data requires also specialised tools [44].

Another important limitation of our study is that we did not utilize the full potential of Nanopore sequencing regarding long read sequencing. By using long reads and tiling amplicon sequencing, it should be possible to sequence the whole gene without the need of amplifying individual exons. While providing the same diagnostic information, this approach would simplify the protocol and reduce the variability in read depth distribution.

Further, prior to clinical application a standardized workflow for sample processing is required.

In the future, in addition to modern bioinformatic data analysis tools, recently announced innovations in nanopores and sequencing chemistry (R10.4 flow cells and Q20+ sequencing chemistry), that increase raw read accuracy, may further improve the performance of Nanopore sequencing for variant identification [45]. Furthermore, they may enable competitive use compared to other

NGS technologies. As mentioned earlier, Nanopore sequencing is especially attractive compared to other technologies like Illumina sequencing, Ion Torrent sequencing or PacBio sequencing due to its fast processing time, lower costs, and ability to generate long reads [45,46].

Summarized, the results of our study show that state-of-the-art Nanopore sequencing in combination with a dedicated data analysis pipeline has a comparable performance to conventional Sanger sequencing for diagnostic SNP genotyping by amplicon sequencing in a clinical setting. Due to continuous technological improvements, after further in-depth clinical validation, this sequencing technique could be applied in clinical genomics and simplify diagnostic workflows in the future.

## Supporting information

**S1 Fig. Screenshot from IGV showing the MEFV region in which the SNP was solely identified by Nanopore sequencing (red box) in two clinical samples.** As Sanger sequencing shows a poor sequence quality at the start and end of a read, this region cannot be sequenced properly by using the routine Sanger sequencing workflow (MF-9-3 = forward sequencing primer Exon 9/10; MF-10-6 = reverse sequencing primer Exon 9/10). By sequencing an additional amplicon, which spans the region containing the variant, it was possible to confirm the transversion in both samples also by Sanger sequencing (MF-10-2 = forward sequencing primer 3' UTR).
(TIF)

**S2 Fig. Electropherograms from the Sanger sequencing runs of an amplicon spanning the relevant region of the 3' UTR.** The transversion from guanine to thymine at genomic position 3293090 is clearly visible in sample 25 (A) and sample 26 (B).
(TIFF)

**S1 Table. Specific primers used for the amplification of the targets within the MEFV gene.**
(DOCX)

**S2 Table. PCR reaction mixes used for amplification of the targets within MEFV.**
(DOCX)

**S3 Table. PCR reaction programs used for the amplification of the targets within the MEFV gene.**
(DOCX)

**S4 Table. Sequencing primers which were used to sequence the individual amplicons by Sanger sequencing.** The final concentration in the reaction mix was 5 μM.
(DOCX)

**S5 Table. Overview of the barcode assignment which was used to sequence the clinical samples on a MinION sequencing device.** 47 samples were sequenced in four individual runs applying two R9.4.1 flow cells.
(DOCX)

## Acknowledgments

The authors would like to thank Oliver Woll for technical support with Sanger sequencing.

## Author Contributions

**Conceptualization:** Jonas Schmidt, Sandro Berghaus, Frithjof Blessing, Josef Blessing, Folker Wenzel.

**Data curation:** Jonas Schmidt, Sandro Berghaus, Holger Herbeck.

**Formal analysis:** Jonas Schmidt.

**Investigation:** Jonas Schmidt.

**Methodology:** Jonas Schmidt.

**Project administration:** Sandro Berghaus, Josef Blessing, Dirk Roggenbuck.

**Resources:** Holger Herbeck.

**Software:** Holger Herbeck.

**Supervision:** Frithjof Blessing, Dirk Roggenbuck, Folker Wenzel.

**Visualization:** Jonas Schmidt.

**Writing – original draft:** Jonas Schmidt.

**Writing – review & editing:** Jonas Schmidt, Frithjof Blessing, Josef Blessing, Peter Schierack, Stefan Rödiger, Dirk Roggenbuck, Folker Wenzel.

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
