## [Decision Letter · Decision Letter 0]

20 Jan 2022

PONE-D-21-38344Genotyping of familial Mediterranean fever gene (MEFV)- single nucleotide polymorphism - comparison of Nanopore with conventional Sanger sequencingPLOS ONE

Dear Dr. Roggenbuck,

Thank you for submitting your manuscript to PLOS ONE. After careful consideration, we feel that it has merit but does not fully meet PLOS ONE’s publication criteria as it currently stands. Therefore, we invite you to submit a revised version of the manuscript that addresses the points raised during the review process.

We look forward to receiving your revised manuscript.

Kind regards,

J Francis Borgio, Ph.D.,

Academic Editor

PLOS ONE

Journal Requirements:

Additional Editor Comments:

Well organized MS, Kindly make sure the consistency in the methods.

Reviewers' comments:

Reviewer's Responses to Questions

**Comments to the Author**

1. Is the manuscript technically sound, and do the data support the conclusions?

Reviewer #1: Yes

Reviewer #2: Partly

2. Has the statistical analysis been performed appropriately and rigorously? 

Reviewer #1: Yes

Reviewer #2: Yes

3. Have the authors made all data underlying the findings in their manuscript fully available?

Reviewer #1: Yes

Reviewer #2: Yes

4. Is the manuscript presented in an intelligible fashion and written in standard English?

Reviewer #1: Yes

Reviewer #2: Yes

5. Review Comments to the Author

Reviewer #1: Entitled: Genotyping of familial Mediterranean fever gene (MEFV) - single nucleotide polymorphism - comparison of Nanopore with conventional Sanger sequencing

Summary: The manuscript is well written; good structured and presents the data in a comprehensive manner. It is a short, straightforward, convincing paper that provides the empirical data to move forward with using portable MinION nanopore sequencer to accurately SNP genotype Familial Mediterranean fever gene (MEFV) by amplicon sequencing using a dedicated and tested bioinformatic and data/analysis pipeline. The approach will be useful for those in the field to begin to adopt MinION sequencing.

Overall Impact: This paper provides the 'bridge' to go from current practice to reliable SNP genotyping by amplicon sequencing using nanopore sequencing technology.

Major Strengths: It is simple in approach and scope, and convincing. It also provides the workflow and software needed to reproduce their analysis.

Major Weaknesses: No major weaknesses.

Line 64. It would be useful to add the value of the error rate and a suitable citation, as it may help readers who are not familiar with how high is the Nanopore sequencing error rate. It might be even better if it is compared to Sanger sequencing, Illumina, PacBio sequencing platforms. Adding one to two sentences would be sufficient.

Line 135-137. Please state how you pooled your samples, how many runs you had and on how many MinION flow cells.

Line 145. The quality and resolution of Figure 1 should be improved. It would be visually nicer to have the same size of boxes for all steps.

Line 186. Figure 2 can be improved; Please improve cDNA labels, they should be uniformly positioned, either in a vertical or skewed fashion. Now they are overlapping and in some places it is not clear which cDNA label comes first (i.e. c.495C>A and c.442G>C). It is also not clear in this Figure how you compare Sanger and Nanopore data. Figure 2 would better suit at Line 202 instead of Figure 4 and can completely be removed from line 186. In that case, you will need to renumber/reorder all figures.

Line 202. Figure 4 is not really informative. It is already clear from the text that both Sanger and Nanopore sequencing identified all 433 SNPs. The graph is not showing or clarifying anything else. Not sure if this Figure is really needed in the main text.

Line 236. I think it should be d) and not e)?

Line 282. Add approximate value for capital costs in Table 2. Here you have a good opportunity to to help readers and show the big difference in costs between Sanger and Nanopore instruments.

Line 299. What other technologies, be more specific (e.g. sequencing technologies, NGS technologies, or similar). Perhaps be again specific to which exactly technologies you are comparing nanopore sequencer to.

Reviewer #2: General commentary:

Summary: This paper assesses the accuracy of SNP genotyping in the coding region of the MEFV gene using the hand-held MinION device. Amplicon sequencing data for patient samples produced via nanopore sequencing were compared to the gold-standard for SNP genotyping (i.e., Sanger sequencing). The resultant data and subsequent analyses provide a bridge between current clinical practices and the accurate detection of disease-associated SNPs throughout the coding region of the MEFV gene using the MinION device to ultimately increase throughput and decrease cost.

Major strengths: This manuscript is straightforward, well-structured and presents the data in a comprehensive manner. Moreover, the data analysis is simple in both approach and scope. If the automated pipeline developed herein is made accessible, this could be easily repeated in a clinical setting (by individuals that may not have extensive knowledge about nanopore data analysis).

Major weaknesses: My main concern is that despite the clinical applicability of the research at hand, the significance of the results obtained are dampened by lack of consistency in the methods utilized (e.g., DNA input for PCR, equimolar amplicon pooling). Given that this study is largely focused on concordance of Sanger and nanopore sequencing, it is critical that the authors confirm the SNP in question using the current gold-standard.

Line-by-line commentary

Line 59: “This sequencing approach offers the advantages of real-time sequencing, ultra-long read length (average read length up to 10 kb), high throughput and low material requirements...”

The authors mention low material requirement as an advantage of ONT sequencing but 7uL of each amplicon were used for Sanger (Line 116) and 10uL for nanopore (Line 127). This is counter to their claim here; can the authors please elaborate?

Line 70: “This can be done either by testing for the most common mutations (targeted mutation analysis) or by sequencing of selected exons.”

It would be helpful if the authors expanded upon the discussion surrounding current diagnostic techniques (e.g., method, number of SNPs assessed in a clinical setting) as well as the technological limitations in the introduction.

Where does NGS (e.g., Illumina) fit in? Has it been used for the diagnosis of MEFV in a clinical setting?

Line 74: “FMF is inherited autosomal recessive and results from point mutations (single substitutions) in the Mediterranean Fever (MEFV) gene.”

Are all SNPs associated with FMF located within the exons of this gene? If not, please provide an explanation as to why the authors chose to focus exclusively on the exons (i.e., diagnostic value).

Line 107: “PCR amplification of the MEFV target regions was performed stepwise in eight different PCR reactions...”

How much DNA was used per reaction? Were samples quantified before amplification? Why or why not? This information is critical to the repeatability and reproducibility of this work.

Lines 115 & 130: Question- Why were 2 different PCR purification methods used prior to Sanger (ExoSAP-IT) and nanopore sequencing (AMpure XP beads)?

Line 134: “The libraries were prepared with an identical number of samples...”

How many samples per flow cell? How did you pick this number? What was the DNA input per sample for library prep and how much of this was pooled prior to sequencing? Again, this information is critical for repeatability and reproducibility.

Line 161: “Once the automated data analysis pipeline was complete...”

Is the automated data analysis pipeline developed here available on github?

Figure 2 & Table S1: Comment: It would be beneficial to include amplicon length.

Table 2: 11 sequencing reactions must be performed to sequence all target regions with Sanger sequencing.

Why are 11 sequencing reactions required when only 8 amplicons were generated via PCR?

Line 249: “...it was possible to sequence the relevant regions of all MEFV exons with a very high read depth.”

What do the results suggest about read depth requirements for accurate SNP genotyping? It is important to mention that this could be achieved by increasing sample per flow cell which would also decrease price per sample (maybe a future direction). However, the authors should also provide the number of samples were multiplexed in the first place.

Line 263: “Nevertheless, because a comprehensive data-base research did not reveal any information about this transversion and we could not confirm its presence by Sanger sequencing...”

While this is a valid explanation, supporting data is required. Would it be possible to confirm via Sanger sequencing using a different set of primers to produce an amplicon centered on the SNP in these two samples? This would significantly strengthen the results obtained and further highlight the advantages of nanopore sequencing.

Line 295: Question: What benefits do R10.4 flowcells offer the region of interest?

Line 271: Comment: A major flaw of this work is failure to harness the long-read capabilities of ONT sequencing platforms. Previous studies have demonstrated that amplification of the entire gene is in fact feasible. This would also bypass the need for multiple PCR reactions, reduce variability in amplicon coverage, and enabling phasing of the SNPs. Would this information provide valuable diagnostic and therapeutic value in a clinical setting? If so, these researchers should have used primers to generate overlapping amplicons that spanned the entire gene in PCR reactions containing the same amount of DNA at minimum. Because long reads are mentioned as an advantage of nanopore sequencing throughout, these points should be included as a limitation in the discussion section.

The development of a standardized workflow for sample processing should also be mentioned as a requirement prior to clinical applicability to bypass these limitations.

6. PLOS authors have the option to publish the peer review history of their article (what does this mean?). If published, this will include your full peer review and any attached files.

Reviewer #1: No

Reviewer #2: No

---

## [Author Response · Author response to Decision Letter 0]

8 Feb 2022

Journal Requirements:

J1: Please ensure that your manuscript meets PLOS ONE's style requirements, including those for file naming. The PLOS ONE style templates can be found at 

Reply J1: We checked the style requirements and adjusted the file naming accordingly.

J2: You indicated that you had ethical approval for your study. In your Methods section, please ensure you have also stated whether you obtained consent from parents or guardians of the minors included in the study or whether the research ethics committee or IRB specifically waived the need for their consent.

Reply J2: We thank the editors for this helpful comment and added the required statement to the Material & methods section.

“For minor patients, written informed consent was obtained from the parents.” 

J3: We note that you have stated that you will provide repository information for your data at acceptance. Should your manuscript be accepted for publication, we will hold it until you provide the relevant accession numbers or DOIs necessary to access your data. If you wish to make changes to your Data Availability statement, please describe these changes in your cover letter and we will update your Data Availability statement to reflect the information you provide.

Reply J3: Our data analysis pipeline as well as the results which were obtained from diagnostic Sanger sequencing are already publicly available from github (https://github.com/j4yo/MEFV-SNP-Genotyping-Pipeline). The Nanopore sequencing data is already uploaded to the European Nucleotide Archive and we can directly make it publicly available under accession number PRJEB49157 upon acceptance. 

Reviewer #1:

R1-1: Line 64. It would be useful to add the value of the error rate and a suitable citation, as it may help readers who are not familiar with how high is the Nanopore sequencing error rate. It might be even better if it is compared to Sanger sequencing, Illumina, PacBio sequencing platforms. Adding one to two sentences would be sufficient.

Reply 1-1: We thank the Reviewer for this helpful comment and added the following sentence to the Introduction after performing a comprehensive literature research on the error rates of different sequencing technologies:

“Although this is a heterogenous measure, which is influenced by different parameters including sequencing instrument, sequencing protocol and sample type, Nanopore sequencing shows a distinct higher error rate (~6%) compared to PacBio sequencing (~1.5%), Illumina sequencing (~0.5%) and conventional Sanger sequencing (~0.001%) [15–19].” 

R1-2: Line 135-137. Please state how you pooled your samples, how many runs you had and on how many MinION flow cells.

Reply 1-2: We appreciate the comment of the reviewer and added the missing information as follows. Further, we added an additional supplementary table (Tab. S5) to clarify the barcode assignment in each run.

“The libraries were prepared with a total of 12 samples per library for each run to ensure a sufficient read count per sample and that the relative proportion of a single sample is comparable (S5 Table). DNA input per sample was 200fmol and 12.5fmol of each barcoded sample were pooled prior to sequencing. Sequencing was performed on a MinION sequencing device (ONT) for 6h using R9.4.1 flow cells (ONT). All samples were sequenced in four different runs using two flow cells. Prior to reuse, the flow cells were purged according to the manufacturer’s protocol using flow cell wash kits (EXP-WSH003) (ONT).” 

R1-3: Line 145. The quality and resolution of Figure 1 should be improved. It would be visually nicer to have the same size of boxes for all steps.

Reply 1-3: We apologize for the poor figure quality. To solve this issue, we increased figure resolution and adjusted the size of the boxes.

R1-4: Line 186. Figure 2 can be improved; Please improve cDNA labels, they should be uniformly positioned, either in a vertical or skewed fashion. Now they are overlapping and in some places it is not clear which cDNA label comes first (i.e. c.495C>A and c.442G>C). It is also not clear in this Figure how you compare Sanger and Nanopore data. Figure 2 would better suit at Line 202 instead of Figure 4 and can completely be removed from line 186. In that case, you will need to renumber/reorder all figures.

Reply 1-4: We thank the reviewer for this helpful comment. To improve the appearance of Figure 2 (now Figure 3) we completely revised the cDNA labels and added coloring to clarify the comparison between Sanger and Nanopore data. Further, we changed the anchoring in the text and reordered the figures.

R1-5: Line 202. Figure 4 is not really informative. It is already clear from the text that both Sanger and Nanopore sequencing identified all 433 SNPs. The graph is not showing or clarifying anything else. Not sure if this Figure is really needed in the main text.

Reply 1-5: We appreciate the comment of the reviewer and removed Figure 4 to increase the clarity and structuredness of the manuscript.

R1-6: Line 236. I think it should be d) and not e)?

Reply 1-6: We thank the reviewer for this fine observation and corrected the table caption.

R1-7: Line 282. Add approximate value for capital costs in Table 2. Here you have a good opportunity to to help readers and show the big difference in costs between Sanger and Nanopore instruments

Reply 1-7: We thank the reviewer for this helpful comment and added the approximate value for capital costs, which include the sequencing device itself, computing units for data analysis as well as software based on available list prices.

R1-8: Line 299. What other technologies, be more specific (e.g. sequencing technologies, NGS technologies, or similar). Perhaps be again specific to which exactly technologies you are comparing nanopore sequencer to.

Reply 1-8: We appreciate the comment of the reviewer and added Illumina sequencing, Ion Torrent sequencing and PacBio sequencing as comparators as well as a suitable citation. 

“As mentioned earlier, Nanopore sequencing is especially attractive compared to other technologies like Illumina sequencing, Ion Torrent sequencing or PacBio sequencing due to its fast processing time, lower costs, and ability to generate long reads. [44,45]”

Reviewer #2:

R2-1: Line 59: “This sequencing approach offers the advantages of real-time sequencing, ultra-long read length (average read length up to 10 kb), high throughput and low material requirements...”

The authors mention low material requirement as an advantage of ONT sequencing but 7uL of each amplicon were used for Sanger (Line 116) and 10uL for nanopore (Line 127). This is counter to their claim here; can the authors please elaborate?

Reply2-1: We appreciate the helpful comment of the reviewer. The 10µl input of each amplicon was chosen to keep the pooling procedure of the amplicons prior to barcoding as simple as possible. In general, the Nanopore sequencing protocol from Oxford Nanopore Technologies for native barcoding requires 100 - 200 fmol of DNA input. Considering a mean amplicon length of 567 bp this equals ~35 - ~70 ng DNA mass which we interpreted as low material requirement. However, to our knowledge for an amplicon length from 500 - 1000 bp Sanger sequencing requires only around 20ng of template. Therefore, to prevent misinterpretation, we removed the low material requirement statement from the introduction.

R2-2: Line 70: “This can be done either by testing for the most common mutations (targeted mutation analysis) or by sequencing of selected exons.” 

It would be helpful if the authors expanded upon the discussion surrounding current diagnostic techniques (e.g., method, number of SNPs assessed in a clinical setting) as well as the technological limitations in the introduction. 

Where does NGS (e.g., Illumina) fit in? Has it been used for the diagnosis of MEFV in a clinical setting?

Reply2-2: We thank the reviewer for this helpful recommendation and add the following text to the introduction:

“According to expert consensus guidelines for the genetic diagnosis of hereditary recurrent fevers a minimum diagnostic screen should include clearly pathogenic variants which are frequently identified in patients [25]. For FMF this incorporates the exons 2, 3, 5 and 10 of MEFV or a set of nine variants [25]. While DNA sequencing is used in most laboratories for variant analysis, targeted approaches can also be applied by using PCR based or reverse-hybridization based assays [25]. However, these targeted approaches as well as conventional Sanger sequencing suffer from the technological limitation that only a comparably small genetic target range can be covered within a single run. To overcome this limitation, NGS can be applied to sequence gene panels including not only MEFV for the diagnosis of FMF but also genes which are associated with other periodic fever syndromes like mevalonate kinase deficiency (MKD, gene MVK), tumor necrosis factor receptor-associated periodic syndrome (TRAPS, gene TNFRSF1A) and cryopyrin-associated periodic syndrome (CAPS, gene NLRP3) [25,26].”

R2-3: Line 74: “FMF is inherited autosomal recessive and results from point mutations (single substitutions) in the Mediterranean Fever (MEFV) gene.”

Are all SNPs associated with FMF located within the exons of this gene? If not, please provide an explanation as to why the authors chose to focus exclusively on the exons (i.e., diagnostic value).

Reply2-3: We thank the reviewer for this helpful comment. According to the current literature, so far MEFV is the only gene which is known to be associated with FMF. However, the complexity of the FMF genetic background can not be described by a single-gene recessive model. Therefore, it can not be excluded that multiple genes and environmental factors are involved. (Ozdogan et al., Presse Med. 2019 Feb;48:e61-e76.) We focused exclusively on the exons since this is a very well established procedure which is recommended by expert committee guidelines for the genetic diagnosis of FMF. (Shinar et al., Ann Rheum Dis. 2012 Oct;71(10):1599-605) 

To clarify this, we corrected the introduction as follows:

“The disease is a clinical diagnosis and mainly characterized by recurrent fever and serositis, with amyloidosis being a severe complication in untreated individuals [22–24]. FMF is considered to be inherited autosomal recessive and is associated with point mutations (single substitutions) in the Mediterranean Fever (MEFV) gene [22,23].

“In this study, to evaluate the clinical performance of current Nanopore sequencing, we applied this sequencing technique in combination with a dedicated data analysis pipeline for SNP genotyping of selected regions of MEFV in 47 patients and validated the results against diagnostic Sanger sequencing as the gold standard method.”

R2-4: Line 107: “PCR amplification of the MEFV target regions was performed stepwise in eight different PCR reactions...”

How much DNA was used per reaction? Were samples quantified before amplification? Why or why not? This information is critical to the repeatability and reproducibility of this work.

Reply2-4: We appreciate this helpful comment of the reviewer. The Prepito NA Body Fluid kit, which was used for DNA isolation from whole blood samples, is very well established in our lab for routine diagnostic protocols. Since whole blood samples show a high yield in DNA isolation from our experience and Sanger sequencing is only a qualitative technique, no DNA quantification step is performed prior to amplification during this protocol. However, based on our experience from previous method validation experiments, the typical yield is around 2.5µg (25 ng/µl in 100µl elution buffer). As stated in supplementary table S2 2.5µl template are used for the PCR reactions which is equivalent to around 60ng DNA per reaction. 

To clarify this, we added the expected yield after DNA isolation to the Material & methods section as well as the template mass to supplementary Table S2. 

R2-5: Lines 115 & 130: Question- Why were 2 different PCR purification methods used prior to Sanger (ExoSAP-IT) and nanopore sequencing (AMpure XP beads)?

Reply2-5: We thank the reviewer for this thoughtful question. Due to German regulations, research lab areas are spatially and logistically separated from areas used for routine diagnostics in our institute. The ExoSAP-IT kits are well established and routinely used for our diagnostic Sanger sequencing runs. AMpure XP beads on the other side are recommended by Oxford Nanopore Technologies for Nanopore sequencing experiments. Since we received the unpurified amplicons as surplus material from routine diagnostics, we used an AMPure XP bead clean-up after pooling because this was the purification method available. 

R2-6: Line 134: “The libraries were prepared with an identical number of samples...”

How many samples per flow cell? How did you pick this number? What was the DNA input per sample for library prep and how much of this was pooled prior to sequencing? Again, this information is critical for repeatability and reproducibility.

Reply2-6: We appreciate the comment of the reviewer. One library was composed of 12 samples. We picked this number to achieve sufficient reads for each sample in any case (Based on a total read count of around 3 million reads per 6h run we calculated 250000 reads per sample). We have planned with this high read count per sample to allow extensive quality filtering during data analysis if needed. The DNA input per sample was 200fmol and 12.5fmol were pooled prior to sequencing. 

To clarify this, we added the following to the Material & methods section:

“The libraries were prepared with a total of 12 samples per library for each run to ensure a sufficient read count per sample and that the relative proportion of a single sample is comparable (S5 Table). DNA input per sample was 200fmol and 12.5fmol of each barcoded sample were pooled prior to sequencing.”

R2-7: Line 161: “Once the automated data analysis pipeline was complete...”

Is the automated data analysis pipeline developed here available on github?

Reply2-7: As stated in the data availability statement the data analysis pipeline is available from github as a Shell script, which can be applied by experienced users (https://github.com/j4yo/MEFV-SNP-Genotyping-Pipeline). 

R2-8: Figure 2 & Table S1: Comment: It would be beneficial to include amplicon length.

Reply2-8: We thank the reviewer for this helpful comment and included the amplicon length in Table S1 as well as a reference in the figure caption of Figure 3 (previously Figure 2).

R2-9: Table 2: 11 sequencing reactions must be performed to sequence all target regions with Sanger sequencing.

Why are 11 sequencing reactions required when only 8 amplicons were generated via PCR?

Reply2-9: We thank the reviewer for this question. To completely resolve the relevant regions and acquire highest diagnostic accuracy the amplicon spanning exon 2 is sequenced in three different sequencing reactions using three different sequencing primers. Further, for the amplicon spanning the relevant regions of exon 9/10 it is necessary to perform two different sequencing reactions with two different primers. To clarify this, we added the following comment to Table 2.

“To archive highest diagnostic accuracy, 11 sequencing reactions must be performed to sequence all target regions with Sanger sequencing, since amplicon 2 and 8 are sequenced in three and two sequencing reactions, respectively.”

R2-10: Line 249: “...it was possible to sequence the relevant regions of all MEFV exons with a very high read depth.”

What do the results suggest about read depth requirements for accurate SNP genotyping? It is important to mention that this could be achieved by increasing sample per flow cell which would also decrease price per sample (maybe a future direction). However, the authors should also provide the number of samples were multiplexed in the first place.

Reply2-10: We thank the reviewer for this helpful comment. Unfortunately, based on our data it is difficult to make a statement on minimum read depth required for accurate SNP genotyping. During initial data analysis we tried subsampling to account for this question. However, due to the inhomogeneous read depth distribution over the different amplicons this did not lead to meaningful results. According to the literature, a read depth of >100x could be sufficient to accurately call SNPs (Leija-Salazar et al., Mol Genet Genomic Med. 2019 Mar;7(3):e564). In our case this would allow a much higher degree of multiplexing.

To clarify this in the manuscript, we added the following text to the Discussion:

“According to Leija-Salazar et al. a read depth of >100x could be sufficient for accurate variant identification by Nanopore sequencing [10]. Such a threshold would remarkably increase the possible degree of multiplexing in our experimental design. However, due to the inhomogeneous read depth distribution between different amplicons we were not able to evaluate this accurately by subsampling of the data.”

R2-11: Line 263: “Nevertheless, because a comprehensive data-base research did not reveal any information about this transversion and we could not confirm its presence by Sanger sequencing...”

While this is a valid explanation, supporting data is required. Would it be possible to confirm via Sanger sequencing using a different set of primers to produce an amplicon centered on the SNP in these two samples? This would significantly strengthen the results obtained and further highlight the advantages of nanopore sequencing.

Reply2-11: We thank the reviewer for this major comment. To confirm the transversion we designed an additional set of primers to produce an amplicon which spans the region of interest and sequenced it by Sanger sequencing. Thereby, we could confirm the transversion in both samples. The primer sequences as well as PCR protocols were added to supplementary Tables S1, S2, S3 and S4. Further, the electropherograms generated by Sanger sequencing were added to the supplementary material (Figure S2) and we updated supplementary Figure S1.

We added the following text to the Results and Discussion section:

“By sequencing an additional amplicon, spanning this region, it was possible to confirm the transversion in both samples also by Sanger sequencing (S2 Fig).” 

“By sequencing an additional amplicon, spanning the relevant region of the 3’ UTR, we were able to confirm the transversion in both patients also by Sanger sequencing. Taking these additional results into account, our data show a complete agreement between Nanopore and Sanger sequencing. Nevertheless, a comprehensive data-base research did not reveal any information about the clinical relevance of this transversion. Since the initial diagnostic Sanger sequencing runs did not identify this variant, the corresponding variant calls were treated as false positive in the calculation of performance measures.”

R2-12: Line 295: Question: What benefits do R10.4 flowcells offer the region of interest?

Reply2-12: R10 nanopores are designed to increase homopolymer performance and thus the consensus accuracy. We assume that it would not offer a large benefit for our region of interest. Further, at the moment this new pore chemistry has the disadvantages of an increased input compared to conventional R9.4.1 flow cells and a decreased output. However, in our opinion it is an important development which might increase the overall performance of Nanopore sequencing in the future. 

R2-13: Line 271: Comment: A major flaw of this work is failure to harness the long-read capabilities of ONT sequencing platforms. Previous studies have demonstrated that amplification of the entire gene is in fact feasible. This would also bypass the need for multiple PCR reactions, reduce variability in amplicon coverage, and enabling phasing of the SNPs. Would this information provide valuable diagnostic and therapeutic value in a clinical setting? If so, these researchers should have used primers to generate overlapping amplicons that spanned the entire gene in PCR reactions containing the same amount of DNA at minimum. Because long reads are mentioned as an advantage of nanopore sequencing throughout, these points should be included as a limitation in the discussion section.

The development of a standardized workflow for sample processing should also be mentioned as a requirement prior to clinical applicability to bypass these limitations.

Reply2-13: We thank the reviewer for this major comment. As mentioned by the reviewer, sequencing of the whole gene by using long reads would simplify the protocol because a single amplification/enrichment step should be sufficient. However, since FMF is a clinical diagnosis that is corroborated by sequencing and only certain exons should be covered during genetic screening as proposed by the expert committee, in our opinion this would not provide a major diagnostic benefit in a clinical setting. 

Furthermore, we were not sure if the accuracy of current Nanopore sequencing would be sufficient for accurate SNP genotyping at all when we designed the study. We therefore decided to use a very well characterized reference to evaluate the performance of Nanopore sequencing in a clinical setting. 

We added the following section to the Discussion of the manuscript:

“Another important limitation of our study is that we did not utilize the full potential of Nanopore sequencing regarding long read sequencing. By using long reads and tiling amplicon sequencing, it should be possible to sequence the whole gene without the need of amplifying individual exons. While providing the same diagnostic information, this approach would simplify the protocol and reduce the variability in read depth distribution. 

Further, prior to clinical application a standardized workflow for sample processing is required.”

---

## [Decision Letter · Decision Letter 1]

7 Mar 2022

Genotyping of familial Mediterranean fever gene (MEFV)- single nucleotide polymorphism - Comparison of Nanopore with conventional Sanger sequencing

PONE-D-21-38344R1

Dear Dr. Roggenbuck,

We’re pleased to inform you that your manuscript has been judged scientifically suitable for publication and will be formally accepted for publication once it meets all outstanding technical requirements.

Kind regards,

J Francis Borgio, Ph.D.,

Academic Editor

PLOS ONE

Additional Editor Comments (optional):

Revised MS can be accepted

Reviewers' comments:

Reviewer's Responses to Questions

**Comments to the Author**

1. If the authors have adequately addressed your comments raised in a previous round of review and you feel that this manuscript is now acceptable for publication, you may indicate that here to bypass the “Comments to the Author” section, enter your conflict of interest statement in the “Confidential to Editor” section, and submit your "Accept" recommendation.

Reviewer #1: All comments have been addressed

Reviewer #2: All comments have been addressed

2. Is the manuscript technically sound, and do the data support the conclusions?

Reviewer #1: Yes

Reviewer #2: Yes

3. Has the statistical analysis been performed appropriately and rigorously? 

Reviewer #1: Yes

Reviewer #2: Yes

4. Have the authors made all data underlying the findings in their manuscript fully available?

Reviewer #1: Yes

Reviewer #2: Yes

5. Is the manuscript presented in an intelligible fashion and written in standard English?

Reviewer #1: Yes

Reviewer #2: Yes

6. Review Comments to the Author

Reviewer #1: Thank you for thoroughly addressing the review comments; I look forward to seeing this work published.

Reviewer #2: (No Response)

7. PLOS authors have the option to publish the peer review history of their article (what does this mean?). If published, this will include your full peer review and any attached files.

Reviewer #1: No

Reviewer #2: No

---

## [Editor Report · Acceptance letter]

9 Mar 2022

PONE-D-21-38344R1 

Genotyping of familial Mediterranean fever gene (*MEFV*)- single nucleotide polymorphism - Comparison of Nanopore with conventional Sanger sequencing 

Dear Dr. Roggenbuck:

I'm pleased to inform you that your manuscript has been deemed suitable for publication in PLOS ONE. Congratulations! Your manuscript is now with our production department. 

Kind regards, 

on behalf of

Dr. J Francis Borgio 

Academic Editor

PLOS ONE